# Long-term Off-Policy Evaluation and Learning

## ABSTRACT

Short- and long-term outcomes of an algorithm often differ, with damaging downstream effects. A known example is a click-bait algorithm, which may increase short-term clicks but damage long-term user engagement. A possible solution to estimate the long-term outcome is to run an online experiment or A/B test for the potential algorithms, but it takes months or even longer to observe the long-term outcomes of interest, making the algorithm selection process unacceptably slow. This work thus studies the problem of feasibly yet accurately estimating the long-term outcome of an algorithm using only the historical and short-term experiment data. Existing approaches to this problem either need a restrictive assumption about the short-term outcomes called *surrogacy* or cannot effectively use short-term outcomes, which is inefficient. Therefore, we propose a new framework called *Long-term Off-Policy Evaluation (LOPE)*, which is based on reward function decomposition. LOPE works under a more relaxed assumption than surrogacy and effectively leverages short-term rewards to provably and substantially reduce the variance. Synthetic experiments show that LOPE outperforms existing approaches particularly when surrogacy is severely violated and the long-term reward is noisy. In addition, real-world experiments on large-scale A/B test data collected on a music streaming platform show that LOPE can estimate the long-term outcome of actual algorithms more accurately than existing feasible methods.

## ACM Reference Format:

Anonymous Author(s). 2023. Long-term Off-Policy Evaluation and Learning. In *Proceedings of ACM Conference (Conference'17)*. ACM, New York, NY, USA, 15 pages. https://doi.org/10.1145/nnnnnnn.nnnnnnn

## 1 INTRODUCTION

The rate of algorithmic developments in online services, such as recommender and search engines, is rapid. These developments range from minor adjustments in feature preprocessing to complex alterations within the algorithm itself. Efficient and reliable measurement of the value of these ideas is crucial to the success of such services [5, 16]. In particular, there is often interest in understanding the long-term outcome or reward (e.g., the annual number of active users or revenue) of these algorithmic changes to make informed decisions regarding algorithm evaluations and selection [10, 37].

The most accurate and intuitive method to achieve this is to run an online experiment (or A/B test) comparing new and baseline algorithms over a sufficiently long period. However, this approach

of running two (or more) candidate algorithms for a long period presents several clear disadvantages:

- The process of algorithm selection becomes extremely slow. If we conduct a year-long online experiment to measure the number of active users after a full year of deployment, we may miss opportunities to test other, better technologies that emerge over the course of that year.
- Running an extended experiment with multiple algorithms can be highly risky, as some candidate algorithms may significantly underperform compared to others, negatively impacting user experience.

Therefore, it is advantageous to develop reliable statistical methods that use only existing historical datasets (collected by baseline algorithms prior to the experiment) and short-term experiments (e.g., several weeks to a month) to accurately estimate the long-term outcome of algorithms, allowing for quicker yet accurate algorithm evaluation and selection [5].

A conventional approach to estimating the long-term outcome of algorithmic changes is through a method known as *long-term causal inference (LCI)* [3, 4, 15, 20, 27]. LCI aims to achieve this by using historical data to infer the causal relationship between short-term surrogate outcomes (such as clicks, likes) and long-term outcomes (such as user activeness indicator a year from now). For this to be valid, LCI necessitates an assumption known as **surrogacy**. This requires that the short-term outcomes hold sufficient information to identify the distribution of the long-term outcome. However, this assumption has been considered restrictive and challenging to satisfy because it demands the presence of sufficient short-term surrogates that enable perfect identification of the long-term outcome [5]. In other words, the connection between short-term and long-term outcomes must remain entirely consistent across all algorithms.

To bypass the restrictive surrogacy assumption, one could potentially apply off-policy evaluation (OPE) techniques [31, 36], such as inverse propensity scoring (IPS) and doubly-robust (DR) methods [12, 28], to historical data for estimating the long-term outcome of algorithms. Unlike LCI, OPE employs action choice probabilities under new and baseline algorithms, eliminating the need for surrogacy. However, typical OPE methods cannot take advantage of short-term rewards, which could be very beneficial as weaker yet less noisy signals particularly when the long-term reward is noisy.

To address the limitations of both LCI and OPE methods, we propose a new framework and method, which we call *Long-term Off-policy Evaluation (LOPE)*. LOPE is a new OPE problem where we aim to estimate the long-term outcome of a new policy, but we can use short-term rewards and short-term experiment data. To solve this new statistical estimation problem efficiently, we develop a new estimator that is based on a decomposition of the expected long-term reward function into surrogate and action effects. The surrogate effect is a component of the long-term reward that can be explained by the observable short-term rewards, while the action effect is the residual term that cannot be captured solely by the short-term surrogates and is also influenced by the specific choice of

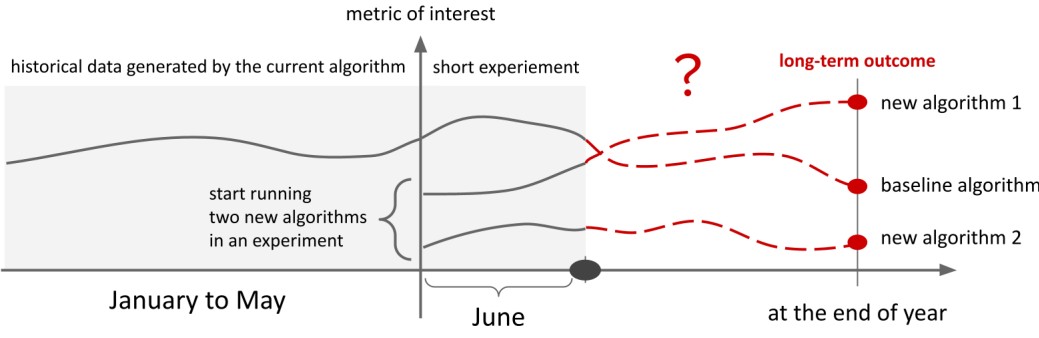

Figure 1: The Statistical Problem of Estimating The Long-term Outcomes Using Historical and Short-term Experiment Data

*Note*: The figure illustrates an example situation where a baseline algorithm was running until May, which generates historical data. An online experiment comparing a baseline and two new algorithms was launched in June, producing short-term experiment data. At the end of June, the baseline algorithm performed the best followed by new algorithm 1. However, at the end of the year, new algorithm 1 will be performing better, but this trend is not yet observed at the end of the short experiment. We aim to infer this long-term outcomes using only available data.

actions or items. The surrogacy assumption of LCI can be viewed as a special case of this reward function decomposition since surrogacy is an assumption that completely ignores the action effect. As LOPE considers a non-zero action effect, it is a strictly more general formulation than LCI. To develop a new estimator for the long-term outcome, we estimate the surrogate effect via importance weights defined using short-term rewards. The action effect is then addressed via a reward regression akin to LCI.

To demonstrate the effectiveness of LOPE formally, we provide a theoretical analysis of its statistical properties showing that (1) it is unbiased under a less restrictive assumption than surrogacy of LCI, and (2) it produces much lower variance than typical OPE by leveraging short-term rewards to estimate the surrogate effect. We also design a new policy learning algorithm to directly optimize the long-term outcome based only on historical data by applying LOPE to policy-gradient estimation. Finally, we perform experiments on synthetic data and real-world data collected from a large-scale music streaming service. The results show that LOPE provides more accurate estimation, algorithm selection, and policy learning in terms of the long-term outcome than LCI and OPE methods, particularly with large reward noise and violation of surrogacy.

## 2 PROBLEM FORMULATION

We first formulate the statistical problem of estimating algorithms' long-term values. Let $x \in \mathcal{X}$ denote the context vector (e.g., user demographics, consumption history, weather) and $a \in \mathcal{A}$ denote a (discrete) action where $a$ indicates a particular product, video, song, or a ranking of them. $w \in \{0, 1\}$ identifies a model or algorithm. For recommender systems applications, $w$ could be seen as a recommender model index where $w = 0$ indicates a baseline model currently running on the system, while $w = 1$ indicates a new model not yet deployed. $\pi_w$ denotes the action distribution conditional on the context vector $x$ (typically called a policy) induced by the model $w$, and thus $\pi_w(a|x)$ indicates the probability of action $a$ being chosen by model $w$ for a given $x$. We also consider two types of rewards, namely short-term and long-term rewards, which are denoted by $s \in \mathcal{S}$ and $r \in [0, r_{\max}]$, respectively. The long-term

reward is the primary metric of interest such as the user active-ness indicator a year from now. Long-term reward is often hard to observe directly since it requires deploying a policy for some long period. The short-term rewards can be multi-dimensional and typically consist of weaker signals such as clicks, conversions, likes, dislikes, and shorter-term user activeness, and are much easier to observe and less noisy than the long-term reward. It is thus crucial to leverage short-term reward observations in situations where the long-term reward is extremely noisy.

We define the long-term value (a measure of effectiveness) of a policy $\pi_w$ induced by model $w$ by the expected long-term reward:

$$V(\pi_w) := \mathbb{E}_{p(x)\pi_w(a|x)p(r|x,a)}[r] = \mathbb{E}_{p(x)\pi_w(a|x)}[q(x,a)], \quad (1)$$

where $p(x)$ is an unknown context distribution and $p(r|x, a)$ is an unknown long-term reward distribution. $q(x, a) := \mathbb{E}_{p(r|x,a)}[r]$ is the expected long-term reward function given context vector $x$ and action $a$. If the long-term reward is defined as a binary indicator of whether a user remains active a year from now, $V(\pi_w)$ is the expected fraction of users still active a year later under policy $\pi_w$. **The main goal of this work is to develop an estimator $\hat{V}$ that can accurately estimate the long-term value of a new model, i.e., $V(\pi_1)$, without running a long-term experiment of $\pi_1$.**

Below, we summarize existing approaches and their limitations.[1]

*Long-term Experiment.* The Long-term Experiment method de-ploys $\pi_1$ for a long period and obtains the following data [9, 17].

$$\mathcal{D}_E := \{r_i\}_{i=1}^{n_E}, \ r \sim p(r \mid \pi_1), \quad (2)$$

where $E$ stands for *Experiment*. $p(r|\pi_1)$ is a marginal distribution of the long-term reward under $\pi_1$.[2] After running a long-term online experiment and obtaining $\mathcal{D}_E$, we can estimate the long-term value of policy $\pi_1$ by the following empirical average estimator.

$$\hat{V}_{\text{AVG}}(\pi_1; \mathcal{D}_E) := \frac{1}{n_E} \sum_{i=1}^{n_E} r_i. \quad (3)$$

---

[1]In Appendix A, we discuss contributions from related work.
[2]$p(r \mid \pi_1) = \int_{x,s} \sum_{a \in \mathcal{A}} p(x)\pi_1(a|x)p(s|x,a)p(r|x,a,s)dxds.$

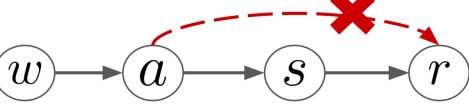

**Figure 2: The Surrogacy Assumption**

*Note*: Grey arrows indicate the existence of causal effect of the tail variable on the head variable. The dashed red arrow is a direct causal effect that is ruled out by Assumption 2.1.

Since we observe the long-term reward $r$ in $\mathcal{D}_E$, the simple estimator in Eq. (3) is unbiased, i.e., $\mathbb{E}_{\mathcal{D}_E}[\hat{V}_{\text{AVG}}(\pi_1; \mathcal{D}_E)] = V(\pi_1)$. However, this method is often infeasible, since we need to deploy the new policy for a long period to observe $r$. Therefore, even though running a long-term experiment enables an accurate estimation of $V(\pi_1)$, it is desirable to develop an alternative statistical method that are much more feasible.

*Long-term Causal Inference (LCI).* LCI enables an estimation of $V(\pi_1)$ without running a long-term experiment [4, 18]. Instead, this approach uses a historical dataset $\mathcal{D}_H$ that contains short-term and long-term rewards, and a short-term experiment dataset $\mathcal{D}_S$ that contains only short-term rewards. $\mathcal{D}_H$ and $\mathcal{D}_S$ are given as

$$\mathcal{D}_H := \{(x_i, s_i, r_i)\}_{i=1}^{n_H} \sim p(x, s, r \mid \pi),$$
$$\mathcal{D}_S := \{(x_i, s_i)\}_{i=1}^{n_S} \sim p(x, s \mid \pi_1),$$

where $\mathcal{D}_H$ is a historical dataset that contains only $(x, s, r)$ and can be generated by an arbitrary policy.[3] $\mathcal{D}_S$ is a short-term experiment dataset generated by running the new policy only for a short period and thus it contains only $(x, s)$.

An important starting point of the LCI approach is the following surrogacy assumption [4, 20]:

**ASSUMPTION 2.1.** *(Surrogacy) The short-term rewards $s$ satisfy surrogacy if $r \perp a \mid x, s$.*

Figure 2 illustrates this assumption, showing that surrogacy requires that every causal effect of the model $w$ or action $a$ on the long-term reward $r$ should be fully mediated by the observable short-term rewards $s$. This enables us to characterize the expected long-term reward function $q(x, a, s)$ by using only the context vector $x$ and short-term rewards $s$ as $\mathbb{E}[r|x, a, s](= q(x, a, s)) = \mathbb{E}[r|x, s](= q(x, s))$. LCI uses this assumption and estimates the long-term value $V(\pi_1)$ as follows:

$$\hat{V}_{\text{LCI}}(\pi_1; \mathcal{D}_S, \mathcal{D}_H) := \frac{1}{n_S} \sum_{i=1}^{n_S} \hat{q}(x_i, s_i; \mathcal{D}_H). \quad (4)$$

This estimator is performed on the short-term experiment data $\mathcal{D}_S$. However, since the long-term reward $r$ is missing in $\mathcal{D}_S$, it leverages a long-term reward predictor $\hat{q}(x_i, s_i; \mathcal{D}_H)$ trained on the historical data $\mathcal{D}_H$ as a proxy. For example, we can obtain $\hat{q}(x_i, s_i; \mathcal{D}_H)$ by performing the following regression problem based on $\mathcal{D}_H$:

$$\hat{q} \in \arg\min_{q' \in Q} \sum_{(x,s,r) \in \mathcal{D}_H} (r - q'(x, s))^2, \quad (5)$$

---

[3]It might be generated by the baseline policy $\pi_0$ or by multiple policies other than $\pi_0$ (i.e., the multiple logger setting [1, 21]). LCI does not care about what policies generated $\mathcal{D}_H$, since it uses the dataset only to infer the connection between $s$ and $r$.

where $Q$ is some class of reward predictors such as random forest or neural networks.

LCI is feasible with only historical and short-term experiment datasets and is justified under surrogacy [3, 4]. Many existing methods to estimate the long-term value follows this high-level framework [5, 18–20]. However, if this assumption is not satisfied, the LCI estimator may produce large bias. It also depends highly on the accuracy of the regression in Eq. (5). Given that Eq. (5) aims to predict the long-term reward, which is often highly sparse and noisy, this regression problem is often very challenging. Besides, LCI does not result in a new learning algorithm that directly optimizes the long-term value $V(\pi)$ since it does not formulate action $a$ nor policy $\pi$.

*Typical Off-Policy Evaluation (OPE).* Another high-level approach to feasibly estimate the long-term value $V(\pi_1)$ is to apply typical OPE estimators on the historical dataeset $\mathcal{D}_H$ [31, 36]. Here we consider a situation where the historical dataset $\mathcal{D}_H$ is generated by a baseline policy $\pi_0$ (called the logging policy in OPE), i.e.,

$$\mathcal{D}_H := \{(x_i, a_i, r_i)\}_{i=1}^{n_H} \sim p(x, a, r \mid \pi_0).$$

Given that the historical dataset $\mathcal{D}_H$ contains long-term reward $r$ observations, we can apply typical estimators such as inverse propensity score (IPS) [28] and doubly robust (DR) [12] as follows:

$$\hat{V}_{\text{IPS}}(\pi_1; \mathcal{D}_H) := \frac{1}{n_H} \sum_{i=1}^{n_H} \frac{\pi_1(a_i|x_i)}{\pi_0(a_i|x_i)} r_i, \quad (6)$$

$$\hat{V}_{\text{DR}}(\pi_1; \mathcal{D}_H) := \frac{1}{n_H} \sum_{i=1}^{n_H} \left\{ \frac{\pi_1(a_i|x_i)}{\pi_0(a_i|x_i)} (r_i - \hat{q}(x_i, a_i)) + \hat{q}(x_i, \pi_1) \right\}, \quad (7)$$

where $\hat{q}(x, a)$ is a predictor of $q(x, a)$ and $\hat{q}(x, \pi_1) := \mathbb{E}_{\pi_1(a|x)}[\hat{q}(x, a)]$. We can obtain $\hat{q}$ by regressing the long-term reward $r$ using context vector $x$ and action $a$ as inputs in the historical dataset. The weight $w(x, a) := \pi_1(a|x)/\pi_0(a|x)$ is called the *importance weight*, and due to this weighting, the IPS and DR estimators achieve unbiased estimation of the long-term value based solely on $\mathcal{D}_H$, i.e.,

$$\mathbb{E}_{\mathcal{D}_H}[\hat{V}_{\text{IPS}}(\pi_1; \mathcal{D}_H)] = \mathbb{E}_{\mathcal{D}_H}[\hat{V}_{\text{DR}}(\pi_1; \mathcal{D}_H)] = V(\pi_1).$$

Despite their unbiasedness, it is well-known that these estimators based on importance weighting are likely to suffer from substantial variance [32–34], which we can see by calculating them as

$$n\mathbb{V}_{\mathcal{D}_H}[\hat{V}_{\text{DR}}(\pi_1; \mathcal{D}_H)] = \mathbb{E}_{p(x)\pi_0(a|x)}[w(x, a)^2 \sigma^2(x, a)]$$
$$+ \mathbb{E}_{p(x)}[\mathbb{V}_{\pi_0(a|x)}[w(x, a)\Delta_{q, \hat{q}}(x, a)]]$$
$$+ \mathbb{V}_{p(x)}[\mathbb{E}_{\pi(a|x)}[q(x, a)]], \quad (8)$$

where $\sigma^2(x, a) := \mathbb{V}[r|x, a]$ and $\Delta_{q, \hat{q}}(x, a) := q(x, a) - \hat{q}(x, a)$. Note that the variance of IPS can be obtained by setting $\hat{q}(x, a) = 0$ by definition. The variance reduction of DR comes from the second term where $\Delta_{q, \hat{q}}(x, a)$ is smaller than $q(x, a)$ if $\hat{q}(x, a)$ is reasonably accurate. However, the first term in the variance can be extremely large for both IPS and DR when the long-term reward is noisy and the weights $w(x, a)$ have a large variation. In particular, the dependence of their variance on the reward noise $\sigma^2(x, a)$ is a serious issue in our long-term value estimation task since the long-term reward $r$ is often substantially noisy. It is thus crucial to be able to leverage short-term rewards $s$ and short-term experiment data

**Table 1: Comparison of the existing and proposed approaches for long-term value estimation**

|  | Is it practically feasible? | Does not it need surrogacy? | Can it utilize short-term rewards? | Can it be turned into a learning algorithm? |
|---|:---:|:---:|:---:|:---:|
| Long-term Experiment | ✗ | ✓ | ✓ | ✗ |
| Long-term CI | ✓ | ✗ | ✓ | ✗ |
| Typical OPE | ✓ | ✓ | ✗ | ✓ |
| **Long-term OPE (ours)** | ✓ | ✓ | ✓ | ✓ |

*Note*: **Long-term Experiment** is often considered infeasible since it needs to run a long-term online experiment to observe the long-term reward $r$ under the new policy $\pi_1$. **Long-term CI** is justified only under the surrogacy assumption (Assumption 2.1). In addition, it does not produce a policy learning method to optimize the expected long-term reward. **Typical OPE** cannot utilize short-term rewards $s$, which is extremely useful as weaker but less noisy signals, particularly when the long-term reward $r$ is noisy, making it suboptimal in variance.

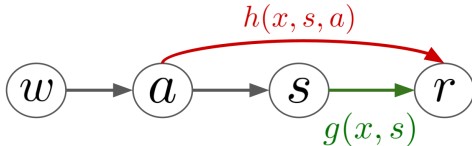

**Figure 3: Reward decomposition employed by LOPE.**

$\mathcal{D}_S$ (if available) to deal with this variance issue of OPE methods. However, there is no existing method in OPE that can exploit short-term rewards to reduce variance.

To achieve this, the following develops a new statistical framework that can take advantage of short-term rewards to provably and substantially reduce the variance of OPE methods while assuming only less restrictive assumptions than LCI.

## 3 LONG-TERM OFF-POLICY EVALUATION

This section proposes our new framework for long-term value estimation called **Long-term Off-Policy Evaluation (LOPE)**, which integrates the strengths of LCI and typical OPE, i.e., LOPE

- can take advantage of short-term rewards and short-term experiment data to deal with noisy long-term rewards and to reduce variance.
- needs only a more relaxed assumption than surrogacy.
- can produce a policy learning algorithm to optimize the long-term value more efficiently.

Table 1 summarizes the comparison among existing methods and our methods. Below, we describe the specific construction of the proposed estimator and its theoretical foundations.

### 3.1 The LOPE Estimator

To construct our method, we begin with introducing the following decomposition of the expected long-term reward function.

$$q(x, a, s) = \underbrace{g(x, s)}_{\text{surrogate effect}} + \underbrace{h(x, a, s)}_{\text{action effect}}, \qquad (9)$$

where the *surrogate effect* $g(x, s)$ is a factor of the expected reward function that is predictable by only the short-term surrogate rewards $s$ while the *action effect* $h(x, a, s)$ is the effect that is not predictable with only the short-term surrogates. This decomposition does not assume anything since we do not posit specific

forms of the $g$ and $h$ functions. Our decomposition can be considered a generalization of the surrogacy assumption of LCI. If we assume that $h(x, a, s) = 0, \forall x, a, s$, then Eq. (9) is reduced to $q(x, a, s) = g(x, s)$, which is equivalently $r \perp a \mid x, s$, recovering Assumption 2.1. Therefore, as long as being based on Eq. (9), the resulting estimator has a weaker condition to work compared to LCI. Eq. (9) also enables us to effectively leverage the short-term rewards to substantially improve typical OPE methods in terms of estimation variance.

Based on this decomposition, we design a new estimator by applying different estimation strategies between the $g$ and $h$ functions. Specifically, our estimator deals with the surrogate effect by applying importance weighting with respect to the short-term reward distributions and with the action effect via a reward regression as

$$\hat{V}_{\text{LOPE}}(\pi_1; \mathcal{D}_H)$$
$$:= \frac{1}{n_H} \sum_{i=1}^{n} \left\{ \frac{\pi_1(s_i|x_i)}{\pi_0(s_i|x_i)} (r_i - \hat{h}(x_i, a_i, s_i)) + \hat{h}(x_i, \pi_1) \right\}, \qquad (10)$$

where $\pi(s|x) = \sum_{a \in \mathcal{A}} \pi(a|x)p(s|x, a)$ is the marginal surrogate distribution induced by policy $\pi$ and $w(x, s) := \pi_1(s|x)/\pi_0(s|x)$ is called the *surrogate importance weight*. We also use $\hat{h}(x, \pi_1) := \mathbb{E}_{\pi_1(a|x)p(s|x,a)}[\hat{h}(x, s, a)]$ in Eq. (10). The first term of LOPE based on the surrogate importance weight aims to estimate the surrogate effect $g(x, s)$ in Eq. (9). In contrast, the second term of LOPE aims to estimate the action effect $h(x, a, s)$ via a reward regression model $\hat{h}$, which can be obtained, for example, by performing $\hat{h} = \arg\min_h \sum_{(x,a,s,r) \in \mathcal{D}_H} (r - h(x, a, s))^2$, on the historical data. At a high-level, our LOPE estimator makes the most of the short-term reward observations to estimate the surrogate effect $g$ while typical OPE does not estimate the $g$ function separately from the other factor of the reward function. In addition, LOPE addresses the action effect by applying a reward predictor $\hat{h}$ while LCI completely ignores the $h$ function by assuming surrogacy.

Below, we demonstrate desirable theoretical properties of LOPE. First, the following shows that LOPE can be unbiased under a new *doubly-robust* style condition.

THEOREM 3.1. *LOPE is unbiased, i.e.,* $\mathbb{E}_{\mathcal{D}_H}[\hat{V}_{\text{LOPE}}(\pi_1; \mathcal{D}_H)] = V(\pi_1)$, *if either of the following holds true:*

(1) *the surrogacy assumption (Assumption 2.1)*
(2) *the conditional pairwise correctness assumption, which requires:* $q(x, a, s) - q(x, b, s) = \hat{h}(x, a, s) - \hat{h}(x, b, s), \forall a, b \in \mathcal{A}, s \in \mathcal{S}$.

Theorem 3.1 provides conditions needed for the unbiasednesss of LOPE. It is unbiased under the surrogacy assumption. However, it can also be unbiased under a new assumption about the reward function estimator ($\hat{h}$) called *conditional pairwise correctness*, which only requires the reward function estimator accurately estimate the relative long-term reward differences between two different actions, $a$ and $b$, conditional on short-term rewards $s$. This assumption is weaker than an accurate estimation of the long-term reward function $q(x, a, s)$.[4] The important point here is that LOPE needs only one of the assumptions to become unbiased, which can be considered a new *doubly-robust* style guarantee.

Next, the following shows that the surrogate importance weight of LOPE can have substantially lower variance than the vanilla importance weight of IPS and DR.

THEOREM 3.2. *The difference in the variance of the surrogate and vanilla importance weights can be represented as follows.*

$$\mathbb{V}_{p(x)\pi_0(a|x)}[w(x, a)] - \mathbb{V}_{p(x)\pi_0(a|x)p(s|x,a)}[w(x, s)]$$
$$= \mathbb{E}_{p(x)\pi_0(s|x)}\left[\mathbb{V}_{\pi_0(a|x,s)}[w(x, a)]\right],$$

*which is always non-negative.*

Theorem 3.2 characterizes the reduction in variance provided by surrogate importance weighting of LOPE. It is worth noting that the variance reduction is characterized by the variance of the vanilla importance weight $\mathbb{V}_{\pi_0(a|x,s)}[w(x, a)]$, which suggests that surrogate importance weighting provides increasingly larger variance reduction when typical weighting has a larger variance. Based on Theorem 3.2, in Appendix B, we also show that the noise-dependent term in the variance of LOPE has substantially lower variance than that in the variance of IPS and DR.

## 3.2 Estimating Surrogate Importance Weights

When using LOPE, we need to estimate the surrogate importance weight $w(x, s)$ from the logged data. We can achieve this by leveraging the following formula (derived via Bayes' rule).

$$w(x, s) = \mathbb{E}_{\pi_0(a|x,s)}[w(x, a)]. \tag{11}$$

Eq. (11) implies that we need an estimate of $\pi_0(a|x, s)$, which we can derive by regressing $a$ on $(x, s)$. We can then estimate $w(x, s)$ by computing $\hat{w}(x, s) = \mathbb{E}_{\hat{\pi}_0(a|x,s)}[w(x, a)]$. This estimation procedure is easy to implement and tractable, even when the short-term reward $s$ is high-dimensional and continuous. It is worth mentioning that, even though LOPE is feasible with only historical data $\mathcal{D}_H$, we can additionally use short-term experiment data $\mathcal{D}_S$ (if available) to estimate $\pi_0(a|x, s)$ more accurately, which is expected to improve estimation of the surrogate effect.

## 3.3 Extension to Policy Learning

Beyond estimation of the long-term value $V(\pi_1)$, we can formulate a problem of learning a new policy to improve the long-term reward using only historical data. We can formulate this **long-term off-policy learning (long-term OPL)** problem as

$$\max_{\theta} V(\pi_\theta)$$

where $\theta \in \mathbb{R}^d$ is a policy parameter and $V(\pi)$ is defined in Eq. (1). A typical approach to solve this learning problem is the policy-based approach, which updates the policy parameter via iterative gradient ascent as $\theta_{t+1} \leftarrow \theta_t + \nabla_\theta V(\pi_\theta)$. Since we do not know the true gradient $\nabla_\theta V(\pi_\theta) = \mathbb{E}_{p(x)\pi_\theta(a|x)}[q(x, a)\nabla_\theta \log \pi_\theta(a \mid x)]$, we need to estimate it from the logged data where we can apply LOPE. In Appendix D, we show how we can easily extend LOPE to estimate the policy gradient $\nabla_\theta V(\pi_\theta)$ to learn a new policy to improve the long-term value based only on the historical data $\mathcal{D}_H$.

The next section empirically demonstrates that LOPE results in a better policy compared to baseline OPL methods of using IPS (Eq. (6)) and DR (Eq. (7)) to estimate the policy gradient $\nabla_\theta V(\pi_\theta)$.

## 4 SYNTHETIC EXPERIMENTS

This section evaluates LOPE on synthetic data to identify the situations where it is particularly appealing to perform policy evaluation, selection, and learning regarding the long-term value $V(\pi_1)$.[5]

### 4.1 Synthetic Data

We create synthetic datasets to compare the estimates to the ground-truth long-term value of policies. We first define 1,000 unique (synthetic) users represented by 10-dimensional feature vectors $x$, which are sampled from the standard normal distribution. We then synthesize the expected reward function given $x$ and $a$ as

$$q(x, a; \lambda) = (1 - \lambda)g(x, f(x, a)) + \lambda h(x, a) \tag{12}$$

where the $g$ and $h$ functions define the surrogate and action effects, and the $f$ function specifies the expected short-term reward function. Appendix D defines the $f$, $g$, and $h$ functions in greater detail. Note that $|\mathcal{A}| = 30$ in our synthetic experiments. $\lambda \in [0, 1]$ is an experiment parameter to control the violation of surrogacy. When $\lambda = 0$, surrogacy is satisfied while a larger value of $\lambda$ increasingly violates the assumption. We use $\lambda = 0.5$ as a default setup throughout the synthetic experiment to compare methods under a moderate violation of surrogacy given that, in practice, it is hard to satisfy and verify with observable data (we vary the value of $\lambda$ in one of our simulations to see its effect on the comparison of methods).

We synthesize the baseline (logging) policy $\pi_0$ by applying the softmax function to the expected reward function $q(x, a)$ as

$$\pi_0(a \mid x; \beta) = \frac{\exp(\beta \cdot q(x, a))}{\sum_{a' \in \mathcal{A}} \exp(\beta \cdot q(x, a'))}, \tag{13}$$

where $\beta$ is a parameter that controls the optimality and entropy of the logging policy, and we use $\beta = 0.5$ as default.

In contrast, we define the new policy $\pi_1$ by applying an epsilon-greedy rule as

$$\pi_1(a \mid x; \epsilon) = (1 - \epsilon) \cdot \mathbb{I}\left\{a = \arg\max_{a' \in \mathcal{A}} q(x, a')\right\} + \frac{\epsilon}{|\mathcal{A}|}, \tag{14}$$

where the noise $\epsilon \in [0, 1]$ controls the quality of $\pi_1$, and we set $\epsilon = 0.1$ as default.

---

[4]This is because, if $q(x, a, s) = \hat{h}(x, a, s)$, conditional pairwise correctness is true, but the satisfaction of conditional pairwise correctness does not necessarily imply $q(x, a, s) = \hat{h}(x, a, s)$.

[5]Note that our experiment code for the synthetic experiment can be found in https://anonymous.4open.science/r/long-term-ope-1A25 and will be made public on Github upon publication.

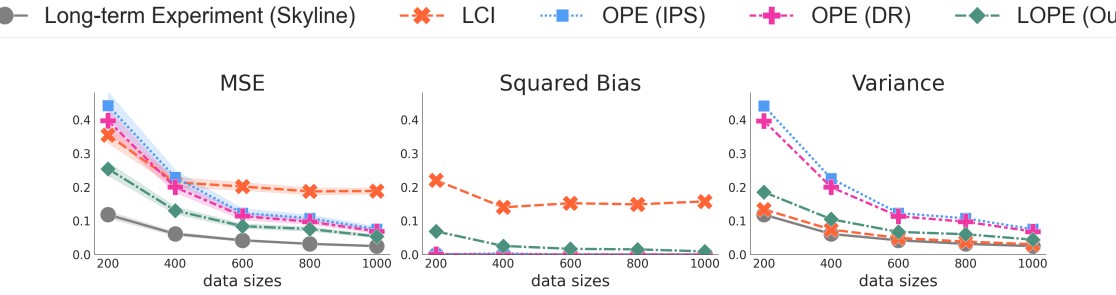

**Figure 4: Comparison of the estimators' MSE, Squared Bias, and Variance with varying data sizes ($n$)**

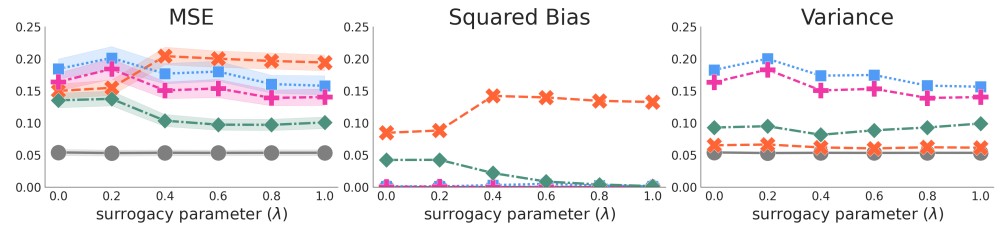

**Figure 5: Comparison of the estimators' MSE, Squared Bias, and Variance with varying violations of surrogacy ($\lambda$)**

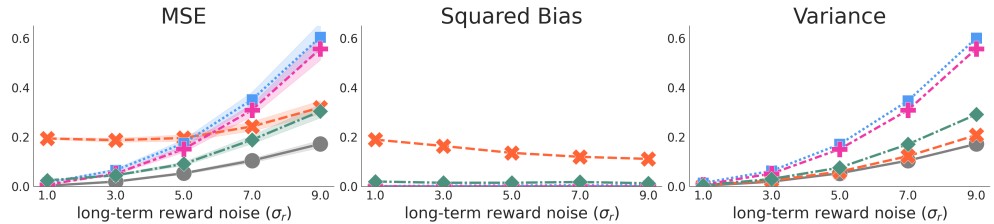

**Figure 6: Comparison of the estimators' MSE, Squared Bias, and Variance with varying reward noise levels ($\sigma_r$)**

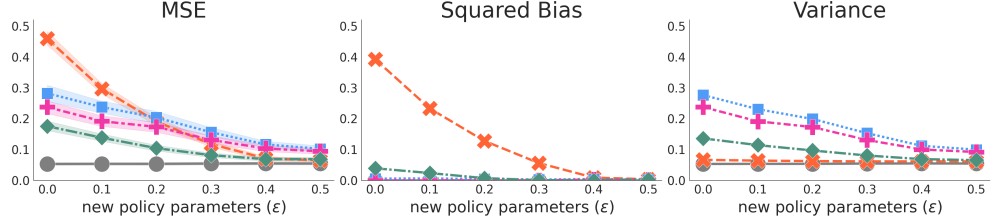

**Figure 7: Comparison of the estimators' MSE, Squared Bias, and Variance with varying new policies ($\epsilon$)**

## 4.2 Results in Policy Evaluation and Selection

Figures 4 to 7 compare the MSE, squared bias, and variance of LOPE (Eq. (10)), LCI (Eq. (4)), IPS (Eq. (6)), and DR (Eq. (7)) against the ground-truth long-term value $V(\pi_1)$. We use the true importance weight $w(x, a)$ for IPS and DR, while we estimate the surrogate importance weight $w(x, s)$ using $\mathcal{D}_H$ and $\mathcal{D}_S$ as in Section 3.2 for LOPE. We also report the accuracy if we could run a long-term

experiment as a skyline reference (grey lines in the figures) showing the best achievable accuracy by feasible methods.[6]

Figure 4 compares the estimators' accuracy regarding long-term value estimation when we vary the size of the historical and short-term experiment data from 200 to 1,000 to compare how sample-efficient each method is. We observe that LOPE provides the lowest MSE among feasible methods in all cases. LOPE is substantially better than OPE methods (IPS and DR), particularly when the data

---

[6]To implement the long-term experiment method, we sample $\mathcal{D}_E = \{r_i\}_{i=1}^{n_E} \sim p(r \mid \pi_1)$ from the same synthetic environment as described in Section 4.1 and compute $\hat{V}_{\text{AVG}}(\pi_1; \mathcal{D}_E)$ as in Eq. (3).

size is small (LOPE achieves 36% reduction in MSE from DR when $n = 200$). This is because LOPE produces substantially lower variance as shown in Theorem 3.2 by effectively combining short-term rewards and long-term reward while OPE methods use only the latter, which is very noisy. In addition, LOPE performs much better than LCI, particularly when the data size is large (LOPE achieves 71% reduction in MSE from LCI when $n = 1,000$), since LOPE has much lower bias. The substantial bias of LCI even with large data sizes is due to its inability to deal with the violation of surrogacy.

Figure 5 reports the estimators' accuracy when we vary the value of $\lambda$ from 0.0 to 1.0 and control the violation of surrogacy (a larger $\lambda$ increases the violation). The MSE of LCI increases with a more severe violation of surrogacy since it produces larger bias under these conditions as shown in the squared bias figure. In contrast, LOPE is not negatively affected by larger violations of the surrogacy assumption, since it is based on a reward function decomposition in Eq. (9) and is free from surrogacy, showing its robustness to the potential violation of this unverifiable assumption.[7]

Figure 6 demonstrates the estimators' accuracy under varying levels of noise of the long-term reward, i.e., $\sigma_r$ from 1.0 to 9.0, to demonstrate how robust each method is to the noisy long-term reward. A larger noise on the long-term reward makes long-term value estimation increasingly difficult and it becomes more important to leverage short-term rewards $s$ as less noisy signals to reduce variance. The left plot of Figure 6 shows that LOPE performs the best in most cases. In particular, when the noise is large, LOPE is much more accurate than IPS and DR since LOPE has substantially lower variance due to the effective use of the short-term rewards (LOPE achieves 45% reduction in MSE from DR when $\sigma_r = 9.0$). In contrast, LOPE is more accurate than LCI when the noise is small, since the bias becomes more dominant in MSE in that case and LOPE generally produces much lower bias due to its less restrictive assumption compared to LCI (LOPE achieves 87% reduction in MSE from LCI when $\sigma_r = 1.0$).

Figure 7 evaluates the estimators' accuracy with varying values of $\epsilon$ from 0.0 to 0.5 to compare how robust each method is to different types of new policies. A smaller value of $\epsilon$ makes the new policy $\pi_1$ increasingly better than the baseline policy $\pi_0$, which makes estimation of $V(\pi_1)$ relatively difficult. The left plot in Figure 7 shows LOPE is particularly effective compared to other methods when the performance difference between $\pi_1$ and $\pi_0$ is large (i.e., smaller $\epsilon$) where estimating $V(\pi_1)$ is challenging. This is because LOPE produces much lower bias than LCI and much lower variance than OPE methods. As $\epsilon$ becomes large, every method performs similarly since the value of the two policies are almost the same. However, given that we are often interested in estimating the value of a new policy that potentially improves the baseline policy, the more accurate estimation by LOPE in such difficult and practical situations is considered appealing.

In addition to the comparison regarding long-term value estimation, we compare the policy selection accuracy of the methods under varying values of $\sigma_r$ (long-term reward noise) from 1.0 to 9.0 and $\epsilon$ (new policy parameter) from 0.0 to 0.5. For every value of $\sigma_r$ and $\epsilon$, the new policy performs better, i.e., $V(\pi_1) > V(\pi_0)$. While

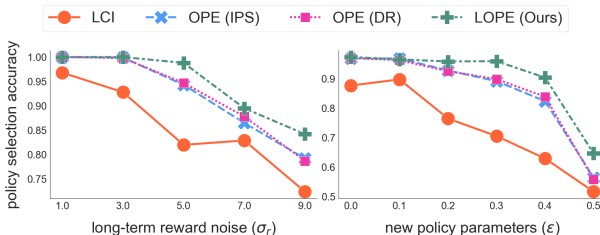

**Figure 8: Comparison of policy selection accuracy of estimators under (left) varying levels of noise on the long-term reward ($\sigma_r$) and (right) new policy parameters ($\epsilon$).**

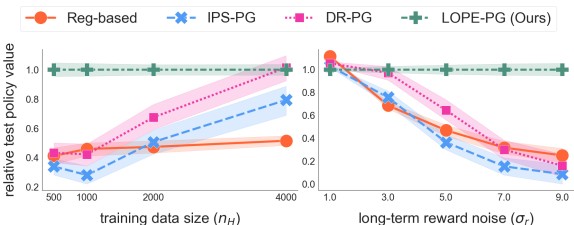

**Figure 9: Comparison of test values $V(\pi)$ of policies learned by each method under (left) varying training data sizes ($n$) and (right) levels of noise on the long-term reward ($\sigma_r$). Values relative to that of LOPE-PG are reported.**

$\sigma_r$ does not have any effect on the values of $\pi_0$ and $\pi_1$, by definition, a larger $\epsilon$ makes the new policy increasingly worse and the difference in the value of the policies smaller, making the policy selection task more challenging.[8] Figure 8 shows the probability that each method identifies the better policy of the two (new and baseline policies) as the accuracy in policy selection. We observe that LOPE becomes particularly effective when the long-term reward noise is large and the value difference between $\pi_1$ and $\pi_0$ is small (i.e., larger $\epsilon$) where identifying the better policy is challenging. Specifically, when $\sigma_r = 1.0$ and $\epsilon = 0.0$, LOPE, IPS, and DR perform perfectly, successfully choosing the better policy without making a mistake. However, as $\sigma_r$ and $\epsilon$ become larger, LOPE becomes more superior to baseline methods. In particular, when $\sigma_r = 9.0$ in the left plot, LOPE is about 85% correct in policy selection while IPS and DR are 79% correct. Moreover, when $\epsilon = 0.5$ in the right plot, LOPE is 65% correct while OPE methods are 55% correct, showing that LOPE enables not only a more accurate evaluation but also a more accurate policy selection regarding long-term value.

## 4.3 Results in Policy Learning

We now compare estimators in terms of their resulting effectiveness in OPL when they are used to estimate the policy gradient $\nabla_\theta V(\pi_\theta)$. Here, we use only historical data $\mathcal{D}_H$ since OPL is a process of learning a new policy, and thus when it is performed, short-term experiment is infeasible. We compare the long-term value of policies learned via IPS, DR, and LOPE (they are called IPS-PG, DR-PG, and LOPE-PG, respectively, where PG stands for Policy Gradient) as well as the regression-based (Reg-based) baseline, which first

---

[7]Note that the MSEs of IPS, DR, and LOPE change with varying values of $\lambda$ even though these methods do not rely on surrogacy. This is because, when we vary the value of $\lambda$, the ground-truth value of the new policy $V(\pi_1)$ also slightly changes.

[8]Specifically, when $\epsilon = 0.0$, $V(\pi_1) = 1.61 > V(\pi_0) = 0.79$, while when $\epsilon = 0.5$, $V(\pi_1) = 0.82 > V(\pi_0) = 0.79$.

**Table 2: Comparison of the estimators' MSE ($\times 10^{-3}$) when estimating the long-term value of each of the three policies. The bold fonts indicate the most accurate method.**

|  | policy #1 | policy #2 | policy #3 |
|---|---|---|---|
| LCI (%Δ from LOPE) | 8.316 (18.8%) | 9.566 (11.0%) | 6.476 (13.3%) |
| IPS (%Δ from LOPE) | 8.474 (21.0%) | 9.735 (13.0%) | 6.614 (15.7%) |
| DR (%Δ from LOPE) | 8.051 (15.0%) | 9.411 (9.2%) | 6.343 (10.9%) |
| LOPE (ours) | **6.999** | **8.615** | **5.715** |

trains a reward predictor $\hat{q}(x, a)$ based on the historical logged data $\mathcal{D}_H \sim \pi_0$, and chooses the best action for each $x$ based on the estimated reward (i.e., a greedy policy based on $\hat{q}$). We use a neural network with 3 hidden layers to parameterize the policy $\pi_\theta$ and obtain $\hat{q}(x, a)$ for LOPE-PG, DR-PG, and Reg-based methods.

Figure 9 compares the long-term value of policies learned by each method in the test set (higher the better) under varying logged data sizes ($n_H$) and levels of noise on the long-term reward ($\sigma_r$). Note that the test policy values achieved by each method relative to that of LOPE-PG are reported (LOPE-PG thus has flat lines).

In the left plot, we see that every method performs more similarly with increased historical data sizes as expected, but particularly when the logged data size is not large (i.e., $n_H = 500, 1000, 2000$), LOPE-PG performs the best (when $n = 500$, LOPE-PG achieved about 60% improvement over DR-PG). This superior performance of LOPE-PG in the small sample regime can be attributed to the fact that it reduces variance in policy-gradient estimation, which leads to a more sample-efficient OPL. Moreover, in the right plot, we can see that LOPE-PG is particularly effective when the long-term reward is noisy. In particular, when $\sigma_r = 9.0$, LOPE-PG outperforms DR-PG by about 80%. This observation is akin to what we observed in Figure 6 regarding OPE accuracy, and empirically demonstrates that the lower-variance policy-gradient estimation achieved by LOPE-PG results in a better long-term value via OPL.

## 5 REAL-WORLD EXPERIMENT

This section demonstrates the effectiveness of LOPE using the logged data collected on a real-world music streaming platform.

We performed a long-term A/B test of three different policies (policy #1, #2, #3), which optimize content recommendations, over a 3-week period in May of 2023. Approximately four million users, chosen at random, were exposed to one of the recommendation policies during the experiment. We also use a historical logged dataset consisting of several million data points collected by a baseline policy prior to the experiment. In these datasets, for each user request, a recommendation policy recommends a content (as an action) such as a playlist, album, and podcast on the user's home page of the platform. The action space consists of the all candidate contents, and there are over thousands contents (i.e., $|\mathcal{A}| > 1,000$) in our application. As a result of such recommendations, several short-term rewards were logged including streams, clicks, likes, and dislikes at week 1 (day 7). We regard streams at week 3 (day 21) as the long-term reward $r$ and define the long-term value of a policy $V(\pi)$ accordingly.

Using these large-scale historical and experiment datasets, we compare the policy evaluation accuracy of LOPE (ours), LCI, IPS,

and DR by their MSE. Note that these methods use only the historical dataset and short-term experiment data to estimate the long-term value of the policies. In the real-world experiment, we never know the ground-truth long-term policy value $V(\pi)$ as in the synthetic experiment, and thus we consider the estimates of the values of the policies estimated based on the full experiment data as the ground-truth[9] and compute the MSE of LCI, IPS, DR, and LOPE (ours) for each of the three different policies.

Table 2 shows the MSE (lower the better) of estimators when estimating the long-term value of three different policies from #1 to #3. The primary observation is that LOPE most accurately replicates the result of the actual long-term experiment for all three policies. LOPE achieves 9.2% ∼ 15.9% reduction in MSE compared to DR (the second best method), providing a further argument about its applicability in the industrial problem. In addition, the LCI method performs similarly to OPE methods in our real-world experiment. This observation is different from most of the synthetic results (Figures 4 to 8). This is because in our real-world experiment, the action space is much larger than that of the synthetic experiment, and thus the variance issue of OPE methods becomes more severe. Thus, the result also implies that the LOPE estimator and its variance reduction property via leveraging short-term rewards (Theorem 3.2) is still effective in this large-scale problem.

## 6 CONCLUSION AND FUTURE WORK

This paper studied the problem of estimating and optimizing the long-term value of an algorithm without running a long-term online experiment. We proposed a new framework called LOPE, which integrates the advantages of Long-term Causal Inference (LCI) and typical Off-policy Evaluation (OPE) methods. LOPE can exploit the useful short-term reward observations via a simple decomposition of the reward function and achieves lower variance than OPE methods under a doubly-robust unbiasedness. LOPE can also be readily extended to a policy learning procedure to optimize the expected long-term value directly via estimating the policy gradient efficiently. Empirical evaluation demonstrated that LOPE enables the most accurate policy estimation and effective policy learning among feasible methods.

Our work also raises several directions for future studies. First, our discussion does not cover how to pre-process the short-term rewards. However, a more refined method for performing representation learning of the short-term rewards may exist to improve the bias and variance of the LOPE estimator. Moreover, our LOPE framework as well as existing LCI and OPE methods assume the distributions of the short- and long-term rewards remain the same in the historical and experiment datasets (often called the comparability assumption [4, 5]). However, there sometimes exist temporal effects in the reward distributions (e.g., users may prefer different types of songs in different seasons). Developing a framework to deal with violations of the comparability assumption would be considered valuable in better capturing the real-world non-stationarity nature of user preferences. Finally, even though LOPE estimates a week 3 metric most accurately in our real-world experiment, it would be valuable to perform a similar experiment regarding even longer-term metrics in the future.

---

[9]It was regarded as the skyline reference (grey lines) in the synthetic experiment.

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

## A  RELATED WORK

This section summarizes related literature regarding two existing approaches, i.e., LCI and typical OPE.

### A.1  Long-term Causal Inference (LCI)

Our work correlates with the literature on long-term causal effect estimation. Utilizing short-term metrics as surrogates to model long-term causal effects is a typical strategy [15, 27]. Early works relied on the surrogacy assumption requiring the short-term surrogates to entirely mediate the long-term outcome. Athey et al. [3, 4] employed multi-dimensional surrogates, which may collectively fulfill the statistical surrogacy criteria even if no individual metric does so independently. Other recent studies have represented surrogates using sequential models [7], and used surrogates for policy optimization, e.g., McDonald et al. [24], Wang et al. [39], Yang et al. [41]. Several theoretical analyses derive the semiparametric efficiency bound in LCI [6, 20] and deal with unobserved confounders [37], which is of independent interest. However, all existing work in LCI relies on surrogacy or related assumptions, which are debatable and unverifiable, causing significant bias under their violations as shown in our simulations. Furthermore, the conventional LCI framework does not formulate an action distribution (policy) induced by an algorithm, thus failing to leverage algorithm similarities. LCI also fails to produce a learning algorithm for direct long-term outcome optimization. To address these limitations, we first provided a more general formulation that considers policy, enabling us to unify LCI and OPE formulations, and combine the strengths of both to produce our LOPE framework. Consequently, LOPE does not rely on surrogacy and is provably more robust to violations of such controversial assumptions. In addition, LOPE can easily be extended to estimate the policy gradient from historical data, facilitating the learning of a new policy, which is not made possible by LCI methods.

### A.2  Off-Policy Evaluation (OPE)

Off-Policy Evaluation (OPE) refers to a statistical estimation problem that estimates the expected reward under a new decision-making policy using only the historical dataset collected under a different policy. It has gained increasing attention in fields ranging from recommender systems to personalized medicine as a safer alternative to online A/B tests, which might be risky, slow, and sometimes even unethical. Among existing OPE estimators, DM and IPS are commonly considered baseline estimators [12, 30, 31, 38]. DM trains a reward prediction model to estimate the policy value. While DM does not produce large variance, it can be highly biased when the reward predictor is inaccurate. In contrast, IPS allows for unbiased estimation under standard identification assumptions but often suffers from high variance due to large importance weights. Doubly Robust (DR) [11, 13, 22] combines DM and IPS to improve variance while remaining unbiased. However, its variance can still be high under large action spaces [2, 8, 26, 32, 33]. As a result, the primary objective of OPE research has been to optimize the bias-variance tradeoff, and numerous estimators have been proposed to address this challenge [23, 25, 34, 35, 40].

Compared to typical OPE methods, which do not assume or leverage short-term rewards and short-term experiment data, our LOPE is designed to harness these valuable additional inputs via a simple reward function decomposition. The use of short-term rewards is particularly crucial when estimating the long-term outcome, as it is often extremely noisy, and leveraging short-term surrogates as less noisy signals can make a significant difference. Broadly, our idea of reward function decomposition is related to a recent line of work on OPE for large action spaces [32, 33], which is based on a different type of reward function decomposition to leverage useful structure in the action space such as action clusters. While they also provide some variance reduction compared to typical OPE, especially when many actions exist, our motivation and formulation for enabling reasonable algorithm evaluations and selections leveraging short-term rewards are fundamentally different.

## B  OMITTED PROOFS

### B.1  Proof of Theorem 3.1

Proof. To show the unbiasedness of LOPE, here we assume that there exists a function $g : X \times S \to \mathbb{R}$ that satisfies the following:

$$\Delta_{q,\hat{q}}(x, a, s) := q(x, a, s) - \hat{q}(x, a, s) = g(x, s), \tag{15}$$

which requires that the estimation error of the reward function estimator $\hat{q}$ should be characterized by some action-independent function $g(x, s)$. In fact, Eq. (15) is satisfied if either of the surrogacy or conditional pairwise correctness is true.

Based on this assumption, the following derives LOPE's unbiasedness. From the linearity of expectation, first we have

$$\mathbb{E}_{\mathcal{D}}[\hat{V}_{\text{LOPE}}(\pi_1; \mathcal{D}_H)] = \mathbb{E}_{p(x)\pi_0(a|x)p(s|x,a)p(r|x,a,s)}[w(x,s)(r - \hat{q}(x,a,s)) + \hat{q}(x,\pi)].$$

Thus, the following calculates only the expectation of $w(x,s)(r - \hat{q}(x,a,s)) + \hat{q}(x,\pi)$.

$$\mathbb{E}_{p(x)\pi_0(a|x)p(s|x,a)p(r|x,a,s)}\left[w(x,s)(r - \hat{q}(x,a,s)) + \hat{q}(x,\pi)\right]$$

$$= \mathbb{E}_{p(x)\pi_0(a|x)p(s|x,a)}\left[w(x,s)(q(x,a,s) - \hat{q}(x,a,s)) + \hat{q}(x,\pi)\right]$$

$$= \mathbb{E}_{p(x)\pi_0(a|x)p(s|x,a)}\left[w(x,s)g(x,s) + \hat{q}(x,\pi)\right] \quad \because \text{Eq. (15)}$$

$$= \mathbb{E}_{p(x)}\left[\hat{q}(x,\pi) + \sum_{a \in \mathcal{A}} \pi_0(a|x) \sum_{s \in \mathcal{S}} p(s|x,a) \frac{\pi(s|x)}{\pi_0(s|x)} g(x,s)\right]$$

$$= \mathbb{E}_{p(x)}\left[\hat{q}(x,\pi) + \sum_{s \in \mathcal{S}} \frac{\pi(s|x)}{\pi_0(s|x)} g(x,s) \sum_{a \in \mathcal{A}} \pi_0(a|x) p(s|x,a)\right]$$

$$= \mathbb{E}_{p(x)}\left[\hat{q}(x,\pi) + \sum_{s \in \mathcal{S}} \frac{\pi(s|x)}{\pi_0(s|x)} g(x,s) \pi_0(s|x)\right]$$

$$= \mathbb{E}_{p(x)\pi(a|x)p(s|x,a)}\left[g(x,s) + \hat{q}(x,a,s)\right]$$

$$= \mathbb{E}_{p(x)\pi(a|x)p(s|x,a)}\left[q(x,a,s)\right] \quad \because \text{Eq. (15)}$$

$$= V(\pi_1)$$

□

## B.2 Proof of Theorem 3.2

PROOF. Since $\mathbb{E}_{p(x)\pi_0(a|x)p(s|x,a)}[w(x,s)] = \mathbb{E}_{p(x)\pi_0(a|x)p(s|x,a)}[w(x,a)] = 1$, the difference in the variance of surrogate and vanilla importance weights is attributed to the difference in their second moment, which is calculated below.

$$\mathbb{V}_{p(x)\pi_0(a|x)p(s|x,a)}[w(x,a)] - \mathbb{V}_{p(x)\pi_0(a|x)p(e|x,a)}[w(x,s)]$$

$$= \mathbb{E}_{p(x)\pi_0(a|x)p(s|x,a)}[w^2(x,a)] - \mathbb{E}_{p(x)\pi_0(a|x)p(e|x,a)}[w^2(x,s)]$$

$$= \mathbb{E}_{p(x)\pi_0(a|x)p(s|x,a)}\left[w^2(x,a) - \left(\mathbb{E}_{\pi_0(a|x,s)}[w(x,a)]\right)^2\right] \quad \because w(x,s) = \mathbb{E}_{\pi_0(a|x,s)}[w(x,a)]$$

$$= \mathbb{E}_{p(x)}\left[\sum_{a \in \mathcal{A}} \pi_0(a|x) \sum_{s \in \mathcal{S}} p(s|x,a) \left(w^2(x,a) - \left(\mathbb{E}_{\pi_0(a|x,s)}[w(x,a)]\right)^2\right)\right]$$

$$= \mathbb{E}_{p(x)}\left[\sum_{a \in \mathcal{A}} \pi_0(a|x) \sum_{s \in \mathcal{S}} \frac{\pi_0(s|x)\pi_0(a|x,s)}{\pi_0(a|x)} \left(w^2(x,a) - \left(\mathbb{E}_{\pi_0(a|x,s)}[w(x,a)]\right)^2\right)\right] \quad \because p(s|x,a) = \frac{\pi_0(s|x)\pi_0(a|x,s)}{\pi_0(a|x)}$$

$$= \mathbb{E}_{p(x)\pi_0(s|x)}\left[\sum_{a \in \mathcal{A}} \pi_0(a|x,s) \left(w^2(x,a) - \left(\mathbb{E}_{\pi_0(a|x,s)}[w(x,a)]\right)^2\right)\right]$$

$$= \mathbb{E}_{p(x)\pi_0(s|x)}\left[\mathbb{V}_{\pi_0(a|x,s)}[w(x,a)]\right]$$

□

## B.3 Additional Theoretical Facts about Variance Comparison

In this subsection, we provide some additional comparisons in the variance of LOPE and DR.

As a warm-up, we first derive the variance of LOPE by applying the law of total variance several times below.

$$n\mathbb{V}_{\mathcal{D}}[\hat{V}_{\text{LOPE}}(\pi_1; \mathcal{D}_H)] = \mathbb{V}_{p(x)\pi_0(a|x)p(s|x,a)p(r|x,a,s)}[w(x,s)(r - \hat{q}(x,a,s)) + \hat{q}(x,\pi)]$$

$$= \mathbb{E}_{p(x)\pi_0(a|x)p(s|x,a)}[\mathbb{V}_{p(r|x,a,s)}[w(x,s)(r - \hat{q}(x,a,s)) + \hat{q}(x,\pi)]]$$
$$+ \mathbb{V}_{p(x)\pi_0(a|x)p(s|x,a)}[\mathbb{E}_{p(r|x,a,s)}[w(x,s)(r - \hat{q}(x,a,s)) + \hat{q}(x,\pi)]]$$

$$= \mathbb{E}_{p(x)\pi_0(a|x)p(s|x,a)}[w^2(x,s)\sigma^2(x,s)]$$
$$+ \mathbb{V}_{p(x)\pi_0(a|x)p(s|x,a)}[w(x,s)(q(x,a,s) - \hat{q}(x,a,s)) + \hat{q}(x,\pi)]]$$

$$= \mathbb{E}_{p(x)\pi_0(a|x)p(s|x,a)}[w^2(x,s)\sigma^2(x,s)]$$
$$+ \mathbb{E}_{p(x)\pi_0(a|x)}[\mathbb{V}_{p(s|x,a)}[w(x,s)\Delta_{q,\hat{q}}(x,a,s) + \hat{q}(x,\pi)]]$$
$$+ \mathbb{V}_{p(x)\pi_0(a|x)}[\mathbb{E}_{p(s|x,a)}[w(x,s)\Delta_{q,\hat{q}}(x,a,s) + \hat{q}(x,\pi)]]$$

$$= \mathbb{E}_{p(x)\pi_0(a|x)p(s|x,a)}[w^2(x,s)\sigma^2(x,s)]$$
$$+ \mathbb{E}_{p(x)\pi_0(a|x)}[\mathbb{V}_{p(s|x,a)}[w(x,s)\Delta_{q,\hat{q}}(x,a,s)]]$$
$$+ \mathbb{E}_{p(x)}[\mathbb{V}_{\pi_0(a|x)}[\mathbb{E}_{p(s|x,a)}[w(x,s)\Delta_{q,\hat{q}}(x,a,s) + \hat{q}(x,\pi)]]]$$
$$+ \mathbb{V}_{p(x)}[\mathbb{E}_{\pi_0(a|x)p(s|x,a)}[w(x,s)\Delta_{q,\hat{q}}(x,a,s) + \hat{q}(x,\pi)]]$$

$$= \mathbb{E}_{p(x)\pi_0(a|x)p(s|x,a)}[w^2(x,s)\sigma^2(x,s)]$$
$$+ \mathbb{E}_{p(x)\pi_0(a|x)}[\mathbb{V}_{p(s|x,a)}[w(x,s)\Delta_{q,\hat{q}}(x,a,s)]]$$
$$+ \mathbb{E}_{p(x)}[\mathbb{V}_{\pi_0(a|x)}[\mathbb{E}_{p(s|x,a)}[w(x,s)\Delta_{q,\hat{q}}(x,a,s)]]]$$
$$+ \mathbb{V}_{p(x)}[\mathbb{E}_{\pi(a|x)}[q(x,a)]]$$

where $\sigma^2(x,s) := \mathbb{V}_{p(r|x,s)}[r]$ and we assume surrogacy (Assumption 2.1) just for the ease of exposition.

Similarly to the variance of DR, which is given in Eq. (19), the critical term in the variance is the first term $\mathbb{E}_{p(x)\pi_0(a|x)p(s|x,a)}[w^2(x,s)\sigma^2(x,s)]$, which depends on the squared importance weight and long-term reward variance. However, we can actually show that the first term in the variance of LOPE can be substantially smaller than that in the variance of DR.

THEOREM B.1. *The difference in the first term of the variance of LOPE and DR can be represented as follows.*

$$\mathbb{E}_{p(x)\pi_0(a|x)p(s|x,a)}[w^2(x,a)\sigma^2(x,s)] - \mathbb{E}_{p(x)\pi_0(a|x)p(s|x,a)}[w^2(x,s)\sigma^2(x,s)] = \mathbb{E}_{p(x)\pi_0(s|x)}[\sigma^2(x,s)\mathbb{V}_{\pi_0(a|x,s)}[w(x,a)]],$$

*which is always non-negative.*

PROOF. This can be proven similarly to Theorem 3.2.

$$\mathbb{E}_{p(x)\pi_0(a|x)p(s|x,a)}[w^2(x,a)\sigma^2(x,s)] - \mathbb{E}_{p(x)\pi_0(a|x)p(e|x,a)}[w^2(x,s)\sigma^2(x,s)]$$

$$= \mathbb{E}_{p(x)\pi_0(a|x)p(s|x,a)}\left[\left(w^2(x,a) - \left(\mathbb{E}_{\pi_0(a|x,s)}[w(x,a)]\right)^2\right)\sigma^2(x,s)\right] \quad \because w(x,s) = \mathbb{E}_{\pi_0(a|x,s)}[w(x,a)]$$

$$= \mathbb{E}_{p(x)}\left[\sum_{a \in \mathcal{A}}\pi_0(a|x)\sum_{s \in \mathcal{S}}p(s|x,a)\left(w^2(x,a) - \left(\mathbb{E}_{\pi_0(a|x,s)}[w(x,a)]\right)^2\right)\sigma^2(x,s)\right]$$

$$= \mathbb{E}_{p(x)}\left[\sum_{a \in \mathcal{A}}\pi_0(a|x)\sum_{s \in \mathcal{S}}\frac{\pi_0(s|x)\pi_0(a|x,s)}{\pi_0(a|x)}\left(w^2(x,a) - \left(\mathbb{E}_{\pi_0(a|x,s)}[w(x,a)]\right)^2\right)\sigma^2(x,s)\right] \quad \because p(s|x,a) = \frac{\pi_0(s|x)\pi_0(a|x,s)}{\pi_0(a|x)}$$

$$= \mathbb{E}_{p(x)\pi_0(s|x)}\left[\sigma^2(x,s)\sum_{a \in \mathcal{A}}\pi_0(a|x,s)\left(w^2(x,a) - \left(\mathbb{E}_{\pi_0(a|x,s)}[w(x,a)]\right)^2\right)\right]$$

$$= \mathbb{E}_{p(x)\pi_0(s|x)}[\sigma^2(x,s)\mathbb{V}_{\pi_0(a|x,s)}[w(x,a)]]$$

□

Theorem B.1 indicates that the difference in the first term of the variance of two estimators can be represented by the product of the noise $\sigma^2(x,s)$ and the variance of the vanilla importance weight $\mathbb{V}_{\pi_0(a|x,s)}[w(x,a)]$. This suggests that the reduction provided by LOPE becomes increasingly large when the long-term reward noise and the variance of the vanilla importance weight are larger. This is particularly desirable in our long-term value estimation problem since the long-term reward noise is likely to be large. Theorem B.1 also justifies the empirical observation in Figure 6 where we observe that LOPE substantially outperforms DR and IPS when the long-term reward is large due to reduced variance.

## B.4 Derivation of Eq. (11) in Section 3.2

$$
\begin{aligned}
w(x,s) &= \frac{\pi(s|x)}{\pi_0(s|x)} \\
&= \frac{\sum_{a \in \mathcal{A}} p(s|x,a) \cdot \pi(a|x)}{\pi_0(s|x)} \\
&= \frac{\pi_0(s|x) \sum_{a \in \mathcal{A}} (\pi_0(a|x,s)/\pi_0(a|x)) \cdot \pi(a|x)}{\pi_0(s|x)} \quad \because p(s|x,a) = \frac{\pi_0(s|x)\pi_0(a|x,s)}{\pi_0(a|x)} \\
&= \sum_{a \in \mathcal{A}} \pi_0(a|x,s) \frac{\pi(a|x)}{\pi_0(a|x)} \\
&= \mathbb{E}_{\pi_0(a|x,s)} \left[ w(x,a) \right]
\end{aligned}
\tag{16}
$$

## C EXTENSION TO LONG-TERM OFF-POLICY LEARNING (LONG-TERM OPL)

Beyond estimation of the long-term value $V(\pi_1)$, we can formulate a problem of learning a new policy to improve the long-term reward using only historical data. Specifically, we can simply formulate this **long-term off-policy learning (long-term OPL)** problem as

$$
\max_\theta V(\pi_\theta)
$$

where $\theta \in \mathbb{R}^d$ is a policy parameter and $V(\pi)$ is defined in Eq. (1). A typical approach to solve this learning problem is the policy-based approach, which updates the policy parameter via iterative gradient ascent as $\theta_{t+1} \leftarrow \theta_t + \nabla_\theta V(\pi_\theta)$. Since we do not know the true gradient $\nabla_\theta V(\pi_\theta)(= \mathbb{E}_{p(x)\pi_\theta(a|x)}[q(x,a)\nabla_\theta \log \pi_\theta(a\,|\,x)]$ via the log-derivative trick), we need to estimate it from historical logged data $\mathcal{D}_H$.

A common way to do so is to apply importance weighting as follows.

$$
\nabla_\theta \widehat{V}_{\text{IPS-PG}}(\pi_\theta; \mathcal{D}_H) := \frac{1}{n} \sum_{i=1}^n \frac{\pi_\theta(a_i\,|\,x_i)}{\pi_0(a_i\,|\,x_i)} r_i \nabla_\theta \log \pi_\theta(a_i\,|\,x_i) = \frac{1}{n} \sum_{i=1}^n w(x_i, a_i) r_i s_\theta(x_i, a_i),
\tag{17}
$$

where $w(x,a) := \pi_\theta(a\,|\,x)/\pi_0(a\,|\,x)$ is the (vanilla) importance weight and $s_\theta(x,a) := \nabla_\theta \log \pi_\theta(a\,|\,x)$ is the policy score function.

The IPS policy gradient in Eq. (17) is unbiased (i.e., $\mathbb{E}[\nabla_\theta \widehat{V}_{\text{IPS-PG}}(\pi_\theta; \mathcal{D}_H)] = \nabla_\theta V(\pi_\theta)$) under the following full support condition.

ASSUMPTION C.1. *(Full Support) The logging policy $\pi_0$ is said to have full support if $\pi_0(a\,|\,x) > 0$ for all $a \in \mathcal{A}$ and $x \in \mathcal{X}$.*

Unfortunately, though, this requirement of full support can be problematic for two reasons. First, violating the requirement can introduce substantial bias [14, 29, 32]. Second, fulfilling the requirement often leads to excessive variance, since $\pi_0(a\,|\,x)$ becomes small. At first glance, *doubly-robust* (DR) estimation may appear helpful for dealing with the variance issue.

$$
\nabla_\theta \widehat{V}_{\text{DR-PG}}(\pi_\theta; \mathcal{D}_H) := \frac{1}{n} \sum_{i=1}^n \left\{ w(x_i, a_i)(r_i - \hat{q}(x_i, a_i)) s_\theta(x_i, a_i) + \mathbb{E}_{\pi_\theta(a|x)}[\hat{q}(x_i, a) s_\theta(x_i, a)] \right\},
\tag{18}
$$

DR incorporates a reward function estimator $\hat{q}(x,a) \approx q(x,a)$ while maintaining unbiasedness under Assumption C.1, and its variance is often lower than that of Eq. (17). However, unless the rewards are close to deterministic and the reward estimates $\hat{q}(x,a)$ are close to perfect, its variance can still be extremely large due to vanilla importance weighting, which leads to inefficient OPL [32]. Specifically, the issue of the IPS and DR policy gradients can be seen by calculating their variance (of the $j$-th dimension and a particular parameter $\theta \in \mathbb{R}^d$) as

$$
\begin{aligned}
n\mathbb{V}_\mathcal{D} \left[ \nabla_\theta \widehat{V}_{\text{DR-PG}}(\pi_\theta; \mathcal{D}_H)^{(j)} \right] = {}& \mathbb{E}_{p(x)\pi_0(a|x)}[(w(x,a) s_\theta^{(j)}(x,a))^2 \sigma^2(x,a)] \\
&+ \mathbb{E}_{p(x)} \left[ \mathbb{V}_{\pi_0(a|x)}[w(x,a)\Delta_{q,\hat{q}}(x,a) s_\theta^{(j)}(x,a)] \right] \\
&+ \mathbb{V}_{p(x)} \left[ \mathbb{E}_{\pi(a|x)}[q(x,a) s_\theta^{(j)}(x,a)] \right],
\end{aligned}
\tag{19}
$$

where $\sigma^2(x,a) := \mathbb{V}[r\,|\,x,a]$ and $\Delta_{q,\hat{q}}(x,a) := q(x,a) - \hat{q}(x,a)$. $s_\theta^{(j)}(x,a)$ is the $j$-th dimension of the score function. Note that the variance of IPS can be obtained by setting $\hat{q}(x,a) = 0$. The variance reduction of DR comes from the second term where $\Delta_{q,\hat{q}}(x,a)$ is likely to be smaller than $q(x,a)$. However, we can also see that the variance contributed by the first term can be extremely large for both IPS and DR when the long-term reward reward $r$ is noisy and the weights $w(x,a)$ take some large values, which occurs when $\pi_\theta$ assigns large probabilities to actions that are less likely under $\pi_0$.

To deal with the likely high variance of typical off-policy policy gradient estimators, here we extend the LOPE estimator to estimate the policy gradient from historical data as below.

$$
\nabla_\theta \widehat{V}_{\text{LOPE-PG}}(\pi_\theta; \mathcal{D}_H) := \frac{1}{n} \sum_{i=1}^n \left\{ w(x_i, s_i)(r_i - \hat{q}(x_i, a_i, s_i)) s_\theta(x_i, a_i) + \mathbb{E}_{\pi_\theta(a|x)}[\hat{q}(x_i, a, s_i) s_\theta(x_i, a)] \right\},
\tag{20}
$$

where $w(x,s) := \pi_\theta(s|x)/\pi_0(s|x)$.

**Table 3: Improvements in MSE provided by LOPE against feasible methods in the synthetic experiment. A smaller value indicates a larger improvement by LOPE. (default experiment parameters: $n = 500$, $\lambda = 0.5$, $\sigma_r = 0.5$, and $\epsilon = 0.1$). * indicates if the difference in MSE is statistically significant based on the Mann–Whitney U test (p=0.05).**

| | $n = 200$ | $n = 1,000$ | $\lambda = 0.0$ | $\lambda = 1.0$ | $\sigma_r = 1.0$ | $\sigma_r = 9.0$ | $\epsilon = 0.0$ | $\epsilon = 0.5$ |
|---|---|---|---|---|---|---|---|---|
| $\mathrm{MSE}(\hat{V}_{\mathrm{LOPE}})/\mathrm{MSE}(\hat{V}_{\mathrm{LCI}})$ | 0.717* | 0.285* | 0.902* | 0.519* | 0.126* | 0.952 | 0.382* | 1.056 |
| $\mathrm{MSE}(\hat{V}_{\mathrm{LOPE}})/\mathrm{MSE}(\hat{V}_{\mathrm{IPS}})$ | 0.575* | 0.712* | 0.734* | 0.638* | 1.588 | 0.503* | 0.622* | 0.674* |
| $\mathrm{MSE}(\hat{V}_{\mathrm{LOPE}})/\mathrm{MSE}(\hat{V}_{\mathrm{DR}})$ | 0.639* | 0.779* | 0.825* | 0.718* | 3.325 | 0.545* | 0.739* | 0.730* |

As corollaries of already shown theorems, we can show the bias-variance advantages of LOPE as a policy gradient estimator below.

COROLLARY C.1. *The LOPE policy-gradient estimator is unbiased, i.e.,*

$$\mathbb{E}_{\mathcal{D}_H}[\nabla_\theta \widehat{V}_{\mathrm{LOPE-PG}}(\pi_\theta; \mathcal{D}_H)] = \nabla_\theta V(\pi_\theta),$$

*if either of the following holds true:*

(1) *the surrogacy assumption (Assumption 2.1)*
(2) *the conditional pairwise correctness, which requires: $q(x, a, s) - q(x, b, s) = \hat{q}(x, a, s) - \hat{q}(x, b, s), \forall a, b \in \mathcal{A}, s \in \mathcal{S}$.*

COROLLARY C.2. *The difference in the first term of the variance of LOPE and DR (for the j-th dimension and a particular parameter $\theta \in \mathbb{R}^d$) can be represented as follows.*

$$\mathbb{E}_{p(x)\pi_0(a|x)p(s|x,a)}[w^2(x, a)\sigma^2(x, s)] - \mathbb{E}_{p(x)\pi_0(a|x)p(s|x,a)}[w^2(x, s)\sigma^2(x, s)] = \mathbb{E}_{p(x)\pi_0(s|x)}\left[\sigma^2(x, s)\mathbb{V}_{\pi_0(a|x,s)}\left[w(x, a)s_\theta^{(j)}(x, a)\right]\right],$$

*which is always non-negative.*

Corollaries C.1 and C.2 are immediate by following the procedures to prove Theorems 3.1 and B.1 provided in the previous section. In particular, these corollaries show that the LOPE policy-gradient estimator has the same doubly-robust style guarantee regarding its unbiasedness. In addition, it has substantially reduced variance compared to the DR policy-gradient estimator when the long-term reward is noisy and the vanilla importance weight has a large variance. Since LOPE reduces the variance in policy-gradient estimation, it is expected to lead to a more sample-efficient policy learning to improve the long-term value, which we will empirically show in the next section.

# D ADDITIONAL EXPERIMENT DETAILS

## D.1 Detailed Setup

This section describes how we define the reward functions to generate synthetic data. Recall that, in the synthetic experiments, we synthesized the expected long-term reward function as

$$q(x, a; \lambda) = (1 - \lambda)g(x, f(x, a)) + \lambda h(x, a), \tag{21}$$

where we use the following functions as $g(\cdot, \cdot)$ (surrogate effect), $f(\cdot, \cdot)$ (expected short-term rewards), and $h(\cdot, \cdot, \cdot)$ (action effect), respectively.

$$g(x, s) = \theta_{g,c(x)}^\top s, \quad f(x, s) = x^\top M_f e_a + \theta_{f,c(x)}^\top x + \theta_{f,a}^\top e_a, \quad h(x, a) = x^\top M_h e_a + \theta_{h,c(x)}^\top x + \theta_{h,a}^\top e_a,$$

where $(M_f, M_h)$, $(\theta_{g,c(x)}, \theta_{f,c(x)}, \theta_{h,c(x)})$, and $(\theta_{g,a}, \theta_{f,a}, \theta_{h,a})$ are parameter matrices or vectors to define the expected reward. These parameters are sampled from a uniform distribution with range $[-1, 1]$. $c(x)$ is a user cluster, which is learned by performing KMeans in the feature space $X$, and different user clusters have different coefficient vectors in our reward functions. We set the default number of user clusters to 3. $e_a$ is a 5-dimensional feature vector of action $a$, which is sampled from a standard normal distribution.

## D.2 Additional Results

This section reports and discusses additional synthetic experiment results regarding varying numbers of user clusters and varying levels of noise on the short-term reward.

First, we compared the robustness of estimators against varying levels of noise on the short-term reward. Figure 10 shows the MSE, squared bias, and variance under different short-term reward noises. From the figure, we can see that LOPE, IPS, and DR are robust and perform similarly even with increased noise on the short-term reward, while LCI worsens when the short-term reward noise becomes larger. This empirical result suggests that LCI more crucially rely on the quality of the short-term reward. In contrast, LOPE relies on short-term rewards to reduce variance from OPE methods, but it is more robust to noisy short-term reward. Second, we compared the accuracy of estimators with varying numbers of user clusters (which affects the synthetic reward function as described in the previous subsection) to confirm the effect of this experiment parameter. Figure 11 shows the MSE, squared bias, and variance of estimators under varying numbers of user clusters. In the figures, we observe that there exist some small fluctuations in the accuracy metrics of every estimator with different numbers of user clusters, since this experiment parameters changes the reward functions and the value of $\pi_1$. However, it is also true that relative comparison of estimators does not change with this experiment parameter.

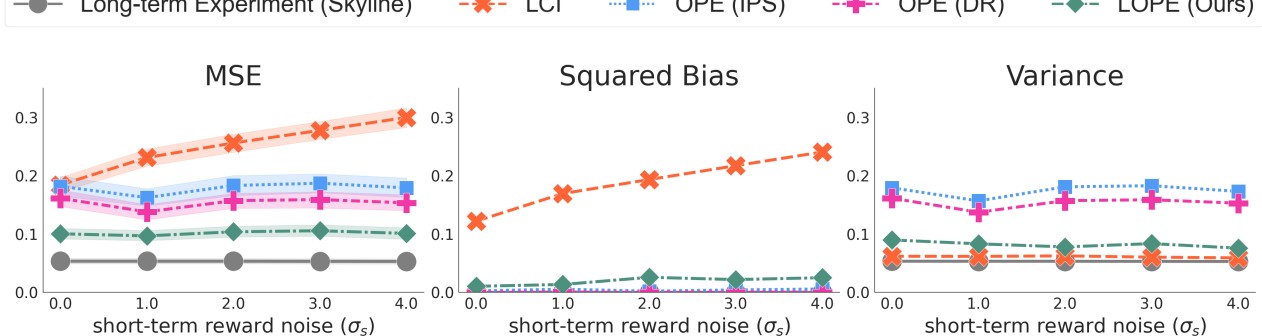

Figure 10: Comparison of the estimators' MSE with varying levels of noise on the short-term rewards ($\sigma_s$)

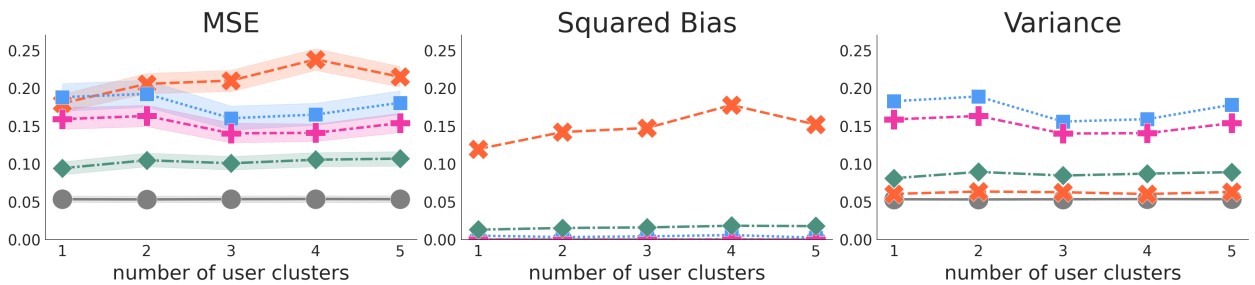

Figure 11: Comparison of the estimators' MSE with varying numbers of user clusters

