# OpenReview forum: "Long-term Off-Policy Evaluation and Learning"
_ACM.org/TheWebConf/2024/Conference — TheWebConf24_

### Official Review · Reviewer_f8km · 2023-11-22

**Novelty:** 5
**Technical Quality:** 6

**Review:**

The goal of this paper is to develop a method to estimate the long-term outcomes from the deployment of a policy without having to run a long-term experiment. A natural first approach would be to consider direct application of off-policy evaluation methods, assuming the availability of historical datasets characterizing some logging policy and its long-term outcomes. This work presents an approach based not only on availability of historical datasets from some logging policy and its long-term outcomes, but also the ability to run short-term experiments with a target policy. The idea is that while it is often not feasible to collect long-term outcomes, it is feasible to run a short-term experiment to log the short-term coutcomes, and these short-term signals can be used to decrease the noise in estimating the long-term outcomes of the target policy relative to using only the historical datasets of a logging policy. An estimator is proposed that that decomposes the expected long term reward into surrogate (component explained by short-term metrics) and action effects (component dependent on action choice). The estimator uses importance weights to estimate the surrogate effect, and the action effect is estimated via reward regression (it is similar to DR style approach). The authors show that this estimator is unbiased under some assumptions that are slightly milder than the surrogacy assumption and has lower variance than typical off-policy evaluation based on only the historical long-term outcomes. They also show that how a policy learning algorithm can be devised based on the estimator. Empirical results comfirm the theoretical results.


This paper is very clearly written and is complete in the sense that it presents an interesting problem that mostly been considered before, along with a solution with some theoretical guarantees and empirical results. I particularly liked the problem formulation, which is certainly a very common problem that comes up in industry. Normally, these seem to be dealt with by the LCI approach from what I have seen, but I agree with the points made by the authors of why this is not also applicable due to the surrogacy assumptions being satisfied. The main question/concern I have is about the surrogate importance weights since the estimator proposed requires these weights and to compute them requires having the distribution p(s|x, a). This distribution would not typically be known. The authors have somewhat addressed this in section 3.2, when talking about estimating the surrogate importance weights by regression, but this is still somewhat problematic given that the theoretical guarantees rely on having the surrogate importance weights and it is hard to know how the estimator will be impacted by error in the estimated surrogate importance weights and the challenge of that regression problem.


Pros
- Well-written paper
- Good, relevant problem formulation
- Nice of mix of theoretical and empirical results

Cons
- Estimator needs a quantity that would not be available. It is a bit unclear how using estimates of the surrogate importance weights impacts the estimator and how feasible it is to estimate the surrogate importance weights.

**Questions:**

See my points about the surrogate importance weights and the need to estimate them. If possible, maybe an experiment can be run to evaluate empirically how error in the surrogate importance weights impacts the estimator.

**Reviewer Confidence:**

3: The reviewer is confident but not certain that the evaluation is correct

**Scope:**

3: The work is somewhat relevant to the Web and to the track, and is of narrow interest to a sub-community

---

### Official Review · Reviewer_ky8t · 2023-11-29

**Novelty:** 6
**Technical Quality:** 6

**Review:**

**Summary**:

This paper proposes a method named "LOPE" to predict long-term reward for a new policy. The previous methods either require the assumption of surrogacy or do not incorporate short-term outcomes. The proposed method addresses the limitations of previous methods and possesses a doubly robust property. Both synthetic experiment and real-world experiment verify the effectiveness of the proposed method.

**Pros**:

- How to combine short-term outcome and long-term outcome is an interesting and important question.
- This paper is well-organized and there are several figures and tables that help reader to better understand this paper.
- The codes are provided to ensure reproducibility.
- Both synthetic experiment and real-world experiment verify the effectiveness of the proposed method.

**Cons**:
- The proposed LOPE differs from the original DR method by incorporating short-term outcomes. As the DR method and its varying have been extensively studied, this limits the novelty of this paper.
- The authors mentioned that predicting long-term reward in Eq. (5) is difficult due to data sparsity and noise. Therefore, it is also hard to obtain a high quality $\hat{h}$ in line 446. It is important to discuss how to learn $\hat{h}$ in this paper.
- This paper should consider more baselines such as the method proposed in [1] in experiments.
- What is the prediction performance when $\pi_1(a_i \mid x_i) / \pi_0(a_i \mid x_i)$ be $\pi_1(s_i \mid x_i) / \pi_0(s_i \mid x_i)$ in Eq. (6)? In addition, what is the prediction performance when only using $\hat{h}$ in line 446 to predict the long-term reward by $\pi_1$?

[1] Lu Cheng, Ruocheng Guo, and Huan Liu. 2021. Long-term effect estimation with surrogate representation. In Proceedings of the 14th ACM International Conference on Web Search and Data Mining. 274–282.

**Questions:**

Please refer to the **Cons** part for questions.

**Reviewer Confidence:**

3: The reviewer is confident but not certain that the evaluation is correct

**Scope:**

3: The work is somewhat relevant to the Web and to the track, and is of narrow interest to a sub-community

---

### Official Review · Reviewer_yz6o · 2023-11-30

**Novelty:** 6
**Technical Quality:** 6

**Review:**

**Strength:**
- *Good Motivation:* The idea of incorporating short-term rewards for a better estimation of long-term outcomes of algorithms while relaxing the surrogacy condition.
- *Proper structure and theoretical analysis:* The paper is easy to follow with its terminology defined clearly and theoretical derivations explained sufficiently.

**Weakness:**
- *Real-world Experiments:* Only one dataset, which is not publicly available and its details are not shared, is being used with no discussion of why existing sequential recommendation datasets or datasets from other domains cannot be used.
- *Synthetic Experiments:* Design choices for generating the synthetic dataset are not justified, and the experiments are done for a limited number of those choices such as a single logging policy, one user sampling distribution, etc. I would at least expect a discussion of why these settings are chosen and why we shouldn't expect considerable change in the results with changing these selections.

---

This paper proposes a solution to the problem of off-policy estimation of the long-term outcomes of an algorithm by incorporating short-term rewards. The paper explains existing methods for feasibly solving this problem:

- The unbiased Typical Off-Policy Evaluation method neglects the weaker but less noisy signal of short-term rewards, resulting in high variance.
- Long-term Causal Inference incorporates short-term outcomes but requires satisfaction of the surrogacy condition to give an unbiased estimation.

The authors propose Long-term OPE, which incorporates short-term rewards to the OPE method to reduce its variance, requiring a more relaxed condition for being unbiased. All motivation discussion and mathematical derivations are properly explained and are rather easy to follow.

Despite the high quality of the paper in those parts, I find the experiments section less satisfying. Different choices in synthetic data generation could be elaborated more. The reason for choosing the logging policy as the softmax function is not immediately clear to me. Since the data is generated synthetically, experimenting with more than one distribution for user sampling, other logging policies, or at least multiple hyperparameters seems to enrich the experimental results section. Furthermore, the explanation about how the actual rewards are generated in Appendix D1 lacks all the required details and selection reasons (e.g., the reason why rewards are defined as this, why clustering is necessary, etc.). Although the experiments repository is attached to the paper, which helps in regeneration of the results, I would strongly suggest more elaboration in the appendix about the synthetic data generation setup.

The thing that I find most concerning about this paper is that the real-world experiments are only performed on one dataset. Also, the dataset is apparently not released, and we don't have sufficient information about it. Even the statistics of the dataset are roughly provided (several million data points, over a thousand actions, etc.) which hinders the reproducibility of the results. I would strongly encourage the authors to provide a discussion about why currently existing datasets are all improper to be used for this paper, and why did they decide to only experiment on a dataset that they cannot share its details. Even if for any reason the existing recommendation datasets could not be used, since the proposed framework is general and there's nothing specific to recommendation, I suppose datasets from other domains could also be useful. Furthermore, the lack of ground-truth rewards and measurable values has forced the authors to compare the estimates of the values provided by policies with the values they provide with having access to the long-term data, an approach that I find questionable. A method that gives constant predictions for the values will obtain a perfect MSE and beat all these methods. All in all, I feel there's a considerable gap between the quality of the real-world experiments and that of the rest of the paper.

**Questions:**

Please find a number of my concerns and questions discussed above. In addition to them, I have two other questions:
1- In Figure 6, although I see the authors' rationale, I observe a decreasing trend in the squared bias of LCI and as expected, its smaller variance compared to LOPI. Considering the fact that here you have 50% of surrogacy violation (I assume lambda being the default 0.5), can we conclude that in extremely noisy settings, LCI's vulnerability to surrogacy violation diminishes, resulting in its better performance compared to LOPI? Concretely, my question is: considering the increasing MSE trend for all methods, will the superiority of LOPI still hold if you further increase the noise level?
2- Is there any particular reason why figures 1 and 3 are provided, but not referred to anywhere in the text?

**Ethics Review Description:**

There are no ethical issues with the paper.

**Reviewer Confidence:**

4: The reviewer is certain that the evaluation is correct and very familiar with the relevant literature

**Scope:**

4: The work is relevant to the Web and to the track, and is of broad interest to the community

---

### Official Review · Reviewer_ZMNH · 2023-12-01

**Novelty:** 6
**Technical Quality:** 5

**Review:**

The paper deals with a problem of how to evaluate the effectiveness of an algorithm in the long term (future) based on limited observational data and some historical stored data of a baseline algorithm. The paper proposes a method to estimate a policy based on the short term experiment and the historical baseline data based on reward decomposition. The statistical method is derived properly and shown to be compatible with previous surrogacy, and a policy method learning is also suggested. Finally the method is evaluated on synthetic experiment designed to highlight the benefit of the suggested method, and on real-world scenario.

**Questions:**

While the derivation and motivation is well established and explained, the experiments part is a not so.
1. in the synthetic experiments, it is unclear what were the durations of each part. There are a lot of details but stating the problem, what is the data set of the historical data and its period, what is the short term experiment and what was considered long term, are missing, or at least hard to uncover from the text.
2. In the real-world experiment, the data is missing and is not present online. The duration of the experiment is only three weeks which seems a little short and I am not sure can be considered as a long term period. The statistics of the process is also not explained, is the distribution steady, is it changing rapidly, this information is crucial to understand the time periods and the validity of the experiments.

**Reviewer Confidence:**

3: The reviewer is confident but not certain that the evaluation is correct

**Scope:**

4: The work is relevant to the Web and to the track, and is of broad interest to the community

---

### Decision · Program_Chairs · 2024-01-22

**Decision:**

Accept

**Comment:**

This paper had a fair amount of reviewer-author interaction with many reviewers increasing their scores during the discussion phase.

 While reviewers raise some concerns about the real-world experiments, experimental design, reproducibility, and whether the long-term claims of the paper can be validated with three weeks of data, it is still nonetheless highly commendable to run real-world experiments at all given the challenge in doing so. That said, there is little I can glean from the necessarily anonymized description other than that LOPE does substantially better and you can attribute it to variance reduction -- though this is only argued in text without any further validation that the improvement is due to reduced variance (not even running a larger action space with the synthetic experiments to validate this claim in a surrogate domain).

 What concerns me a bit more is the synthetic experimentation, where the authors had much more control in experimental design. Looking at the existing experimentation in the main paper and Appendix, I appreciate the controlled research questions being asked about the impact of the various parameters. But I share the reviewers' concerns that this synthetic setup has no motivating use case and only one choice of reward function, baseline policy, and (parameterized) target policy, which fails to fully explore the performance of the proposed algorithm under a wide range of conditions that would be expected in deployment.

 > However, this is not feasible due to the infinite number of potential experimental setups.

 To be fair, the reviewers are not asking for an infinite number of experiments, but some have asked for better practical motivation of the setup and more exploration of reward / baseline policy / target policy choices, which I think is very reasonable.

 Overall, the paper makes a solid technical contribution and has encouraging synthetic and real-world experimental results. For me though, a key weakness is that the authors could have explored more variations of their synthetic experimental setup to better comparatively stress-test their approach under different practically motivated settings.

 As one final remark to the authors, I want to reiterate a revision request of reviewer f8km that would align the theoretical analysis more with the practical implementation of the algorithm:

 > I think it would be worth deriving the bias of the estimator with the estimated surrogate importance weight as you described